# Low-Temperature High-Efficiency Preparation of TiB_2_ Micro-Platelets *via* Boro/Carbothermal Reduction in Microwave Heated Molten Salt

**DOI:** 10.3390/ma12162555

**Published:** 2019-08-11

**Authors:** Jie Liu, Jianghao Liu, Yuan Zeng, Haijun Zhang, Zhi Li

**Affiliations:** The State Key Laboratory of Refractories and Metallurgy, Wuhan University of Science and Technology, Wuhan 430081, China

**Keywords:** TiB_2_, hexagonal microplatelet, microwave heating, molten-salt synthesis

## Abstract

A molten-salt and microwave co-facilitated boro/carbothermal reduction methodology was developed for low temperature high-efficiency synthesis of TiB_2_ powders. By using relatively inexpensive titanium oxide (TiO_2_), boron carbide (B_4_C) and amorphous carbon (C) as raw materials, single-phase TiB_2_ powders were prepared after 60 min at as low as 1150 °C or after only 20 min at 1200 °C. Such synthesis conditions were remarkably milder than those required by the conventional reduction routes using the identical reducing agent. As-synthesized TiB_2_ powders exhibited single-crystalline nature and well-grown hexagonal-platelet-like morphology. The achievement of low temperature high-efficiency preparation of high-quality TiB_2_ microplatelets in the present work was mainly attributable to the synergistic effects of molten-salt medium and microwave heating.

## 1. Introduction

As one of the most important ultra-high temperature ceramics (UHTCs), titanium diboride (TiB_2_) exhibits many useful properties, including a high melting point (3225 °C), low density (4.52 g∙cm^−3^), good corrosion resistance, and high hardness (25–35 GPa) [1,2,3,4]. It is thus regarded as a candidate material that is potentially applicable to versatile structural-applications, e.g., in armors, high-temperature wear-resistant parts, cutting tools, and neutron absorbers [5,6,7]. Currently, the most popular method to prepare TiB_2_-based UHTCs is high temperature consolidation of constituent powders, in which case the synthesis process and the quality of raw material powders (in particular, TiB_2_ powder) are key in optimization of the properties and performance of the final products.

The main synthesis techniques of TiB_2_ powders developed to date can be categorized into three types, including self-propagating high-temperature synthesis (SHS) [8], carbo/boro/metallothermic reduction [9,10,11,12,13,14,15,16,17,18,19,20,21] and sol-gel processing [22,23]. Owing to its relatively low material cost and good processing controllability, the reduction methodology has been and is still being used extensively. For instance, Moradi et al. [13] synthesized TiB_2_ powders from TiO_2_, B_2_O_3_ and graphite *via* carbothermal reduction at 1380 °C for 60 min. Yu et al. [12] prepared TiB_2_ powders by firing powder mixtures of TiO_2_, B_4_C and carbon black at 1600 °C for 30 min. Nevertheless, it is still a great challenge to moderate the synthesis conditions of TiB_2_ and the overall production cost.

To address the aforementioned issues, several advanced strategies, e.g., molten-salt medium and microwave heating, were employed to modify the conventional boro/carbothermal reduction method (BCTR) in the synthesis of TiB_2_ [14,24]. As is well documented [17,18], molten-salt, as a liquid medium, exhibits several important merits in materials synthesis. Apart from reducing the synthesis temperature and accelerating the reaction process, it facilitates diffusive mass transfer and self-assembly processes and thus the formation of highly-crystalline product with intrinsically-anisotropic morphology [25]. On the other hand, microwave heating is capable of selectively heating the material with good microwave absorbability, thus additionally reducing the synthesis temperature and significantly accelerating the overall reaction process. More importantly, it is also effective in inducing the nucleation and therefore accelerating the growth process and enhancing the crystalline degree of synthesized crystals [26,27,28]. Consequently, the introduction of molten-salt medium combined with microwave heating showed crucial positive effects on the overall synthesis process, as well as on the crystalline degree and morphology of the powder product [29,30].

In this work, such a molten-salt and microwave heating co-facilitated boro/carbothermal reduction method (MSM-BCTR), was adopted to prepare TiB_2_ powders with high crystalline degree and anisotropic morphology by using inexpensive TiO_2_, B_4_C and amorphous C as raw materials, while avoiding using any costly B or active metal (Mg or Al) reducing agents. The influence of processing parameters including temperature, dwelling time, microwave/salt medium conditions and the amount of reducing agent on the phase composition and microstructure of powder product was systematically investigated, so as to confirm the superiority of MSM-BCTR over the conventional methods in the preparation of high-quality TiB_2_ powders as well as a broad range of UHTCs.

## 2. Experimental Procedures

### 2.1. Raw Materials

Amorphous carbon was purchased from Sinopharm chem. Co. Ltd (Shanghai, China), boron carbide (B_4_C, purity > 95.0 wt%) was purchased from Mudanjiang Jingangzuan Boron-Carbide Co. Ltd. (Mudanjiang, China). titanium oxide (TiO_2_, rutile phase, purity > 99.9 wt%), sodium chloride (NaCl, purity > 99.9 wt%) and potassium chloride (KCl, purity > 99.9 wt%) were purchased from Bodi Chem. Co. Ltd. (Tianjin, China).
2TiO_2_(s) + B_4_C(s) + 3C(s) = 2TiB_2_(s) + 4CO(g)(1)

B_4_C took part in the overall synthesis reaction (Reaction 1), as both boron source and reducing agent. To compensate for the volatilization loss of boron species at high temperature [30], the amount of B_4_C was slightly higher than the stoichiometric amount indicated by Reaction 1. To minimize the residual carbon in the final powder product, the stoichiometric amount of carbon was used in all the cases. The batch compositions, along with processing conditions, are listed in Table 1.

### 2.2. Methodologies

The reactants and salts (NaCl/KCl) in a weight ratio of 1:2 were mixed manually and then contained in a crucible of 10 mm in diameter and 15 mm in height. The crucible was placed in a columnar saggar, which was subsequently positioned in a microwave furnace (HAMiLab-V3000, Changsha Longtech Co. Ltd. Changsha, China) protected by flowing argon atmosphere. Green silicon carbide (SiC) powder was used to fill the free space of saggar to assist with microwave heating of the powder mixture. The furnace temperature was monitored by an infrared thermometer (Yongtai, Xi’an, China). In a typical MSM-BCTR process, the furnace was firstly heated at a constant rate of 10 °C/min to a pre-determined temperature within 1150–1200 °C, and then held for 0–60 min before cooling naturally to the ambient temperature. The reacted powders were repeatedly rinsed with hot deionized water (80 °C) to remove the residual salt, followed by overnight drying in a vacuum oven at 80 °C.

Phases in raw materials and product samples were analyzed by an X-ray diffractometer (XRD, PANalytical, Hillsboro, The Netherlands) with CuK_α_ radiation (λ = 0.1542 nm) at a scan rate of 5°(2θ)/min. ICDD cards of No.00-035-0741, No.03-065-8805, No.03-035-0798 and No.03-065-1118 were used to identify the crystallographic phases of TiB_2_, TiC, B_4_C and TiO_2_, respectively. Morphologies and sizes of TiB_2_ product powders were examined using a field-emission scanning electron microscope (FE-SEM, 15 kV, Nova400NanoSEM, PHILIPS, Amsterdam, The Netherlands) equipped with an energy dispersive spectrometer (EDS, IET 200, Oxford, UK), and a transmission electron microscope (TEM, 200 kV, JEM-2100UHRSTEM, JEOL, Tokyo, Japan). For TEM characterization, the as-obtained powders were firstly ultrasonically dispersed in ethanol for 30 min, followed by natural seasoning in the air.

## 3. Results and Discussion

Figure 1 presents X-ray diffraction (XRD) patterns of samples prepared by MSM-BCTR at 1150 °C for 20 and 60 min, respectively. In the case of 20 min (sample MSMBC-1), only TiB_2_ and TiC (formed respectively from Reaction 1 and Reaction 2) but no other phases were detected, indicating the complete reduction of the TiO_2_ precursor *via* Reaction 2 and incompletion of Reaction 3. Interestingly, upon just extending the dwelling time at 1150 °C from 20 to 60 min, TiC disappeared and only TiB_2_ was formed (sample MSMBC-2), indicating the high efficiency of the conversion reaction from TiC to TiB_2_ (Reaction 3), and the completion of the overall reaction (Reaction 1) after a short time period at such a low firing temperature.
TiO_2_(s) + 3C(s) = TiC(s) + 2CO(g)(2)
2TiC(s) + B_4_C(s) = 2TiB_2_(s) + 3C(s)(3)

Figure 2 further shows XRD patterns of samples after MSM-BCTR at a slightly higher temperature of 1200 °C. In the case of zero dwelling time (sample MSMBC-3), TiB_2_ was already formed as the main crystalline phase, although certain amounts of reactants (TiO_2_ and B_4_C) still remained along with minor intermediate phase of TiC. Despite this, upon slightly extending the dwelling time to 20 min (sample MSMBC-4), other than the desired TiB_2_, all other impurity phases disappeared completely. Comparing Figure 1 with Figure 2 suggests that such a small enhancement in firing temperature was effective in accelerating not only the direct formation of TiB_2_ but also the conversion from TiC to TiB_2_ (Reactions 1–3). It was worth noting that the synthesis conditions (1150 °C/60 min and 1200 °C/20 min) in the present work were considerably milder than in the cases of conventional BCTR (up to 1600 °C for several hours) using the identical type of reducing agent [12], and the molten-salt-solely-assisted BCTR (firing temperature of 1000 °C and dwelling time of 4 h) using even much more expensive magnesium (Mg) as a reducing agent [16]. These results demonstrated that the present MSM-BCTR method was promising for low-cost, energy-saving, and high-efficiency preparation of TiB_2_ powders, which should be attributed to the combined effect of molten-salt medium and microwave heating, as described in the “Introduction” section and discussed in our previous papers [29,30].

For comparison and further confirming the effectiveness of microwave heating and molten-salt medium on the synthesis of TiB_2_, samples with the same batch compositions were also fired at as-optimized temperature conditions (1200 °C/20 min), using solely microwave heating or salt medium. As shown in Figure 3, in the case without using microwave (sample MSMBC-5), TiO_2_ remained as the dominant phase and no TiB_2_ was detected, indicating the crucial role played by microwave heating in the TiB_2_ formation (*via* Reaction 1–3). On the other hand, due to microwave-assisted BCTR under identical processing conditions, the XRD pattern of sample (sample MSMBC-6) shows the peaks of TiB_2_, B_4_C and TiBO_3_, in which the intensities of TiB_2_ peaks were obviously higher than that of others, indicating the higher crystallization degree of the TiB_2_ phase. Apart from TiB_2_, un-reacted B_4_C were still present along with the intermediate phase of TiBO_3_ which was formed *via* Reaction 4 but tended to be further converted into TiB_2_ (*via* Reaction 5) [13]. The above comparison verified the respective role played by microwave heating and molten-salt medium in the low-temperature rapid synthesis of TiB_2_.
2TiO_2_(s) + B_2_O_3_(g) + C(s) = 2TiBO_3_(s) + CO(g)(4)
2TiBO_3_(s) + B_2_O_3_(g) + 9C(s) = 2TiB_2_(s) + 9CO(g)(5)

In addition to firing temperature and dwelling time, the amount of B_4_C significantly affected the synthesis process and the quality of the product powder. Specifically, the use of inadequate amount of B_4_C would hinder the desirable reduction reaction due to lack of reducing atmosphere, whereas the use of excessive B_4_C would leave high levels of impurities (mainly amorphous C), seriously deteriorating the purity and sinterability of the product powder [9]. Figure 4, as an example, compares phase compositions and reaction extents of the samples initially containing respectively 60 mol% and 40 mol% excess B_4_C, after 20 min MSM-BCTR at 1200 °C. In the case of the former (MSMBC-4), apart from TiB_2_, neither the residual reducing agent nor an impurity phase was detected. However, in the case of the latter (MSMBC-7), despite the presence of the main phase of TiB_2_ and the absence of reducing agent, two diffraction peaks of TiO_2_ with low intensities were still visible, which demonstrated the importance of using appropriately excess amount of reducing agent for ensuring the overall synthetic reactions (Reactions 1–3) to go to completion. It should be noted that, in this process, the requirement of excess boron source was mainly attributable to the evaporation loss of the intermediate product of B_2_O_3_. Specifically, owing to its low melting point and high volatility, B_2_O_3_, as the product of borothermal reduction reaction (Reaction 6) would suffer from significant evaporation loss even at a low temperature range of 900–1100 °C, thus resulting in incompletion of the subsequent carbothermal reduction reaction (Reaction 7) for synthesizing TiB_2_ [21]. Based on the comparison result, it could be concluded that, under the present MSM-BCTR conditions, 60 mol% excessive B_4_C should be used for synthesizing phase pure TiB_2_ powder.
7TiO_2_(s) + 5B_4_C(s) = 7TiB_2_(s) +3B_2_O_3_(s) + 5CO(g)(6)
TiO_2_(s) + B_2_O_3_(s) + 5C(s) = TiB_2_(s) + 5CO(g)(7)

Figure 5 shows microstructure and phase morphology of MSMBC-4, along with the corresponding EDS results. A small amount of monodispersed well-defined hexagonal microplatelets (Figure 5a) coexisted with a large quantity of agglomerates comprising dozens of interlaced micro-platelets (Figure 5b). The corresponding EDS results (presented in Figure 5c–f) revealed that only Ti and B were detected and homogeneously distributed in the microplatelets, confirming that they were TiB_2._ Moreover, except for TiB_2_, no impurity phases (TiO_2_, B_4_C and carbon) were detected, further confirming the high purity of the microplatelets. And the highly-crystallized hexagonal TiB_2_ micro-platelets generally had micron-scale lengths (average length of 4.2 µm) and submicron-scale thicknesses (average thickness of 0.6 µm) with narrow size distributions. Such well-grown hexagonal TiB_2_ microplatelets looked similar to those formed *via* the conventional reduction routes using the identical raw materials, but the synthesis conditions in the case of the latter were much harsher (1600 °C/30 min) [12]. 

TEM (Figure 6a) further reveals the high crystalline extent of as-prepared TiB_2_ particles and their hexagonal plate-like morphology. SAED (Figure 5b) taken from a well-defined TiB_2_ microplatelet (indicated by the red circle in Figure 5a) verifies its single-crystalline nature, and its anisotropic structure formed from the preferential growth along the [010] direction. The high-resolution TEM image shown in Figure 5c further reveals that the hexagonal micro-platelet had well-aligned lattice fringes with a constant interplanar spacing of 0.262 nm, which was in good agreement with the (100) interplanar distance of TiB_2_ crystal [24]. On the basis of these results, it could be considered that the TiB_2_ powders prepared *via* MSM-BCTR exhibited single-crystalline nature, uniform hexagonal plate-like morphology as well as great potential in reinforcing their bulk counterparts. Such results and the much reduced synthesis temperature and dwelling time described and discussed above, were mainly attributed to the synergistic effects of microwave heating and molten salt medium.

## 4. Conclusions

Phase pure TiB_2_ microplatelets were successfully prepared from inexpensive raw material powders of TiO_2_, B_4_C and C *via* MSM-BCTR treatments in KCl-NaCl medium at either 1150 °C/60 min or 1200 °C/20 min. Such synthetic conditions were considerably milder than that required by the conventional BCTR routes using the identical reducing agent. As-synthesized TiB_2_ powders were single-crystalline, and exhibited well-defined hexagonal plate-like morphology with great potential in toughening their bulk counterparts. The achievement in high-efficiency, energy-saving and low-cost preparation of high-quality TiB_2_ microplatelets was attributable to the synergistic effects of microwave heating and molten-salt medium.

## Figures and Tables

**Figure 1 materials-12-02555-f001:**
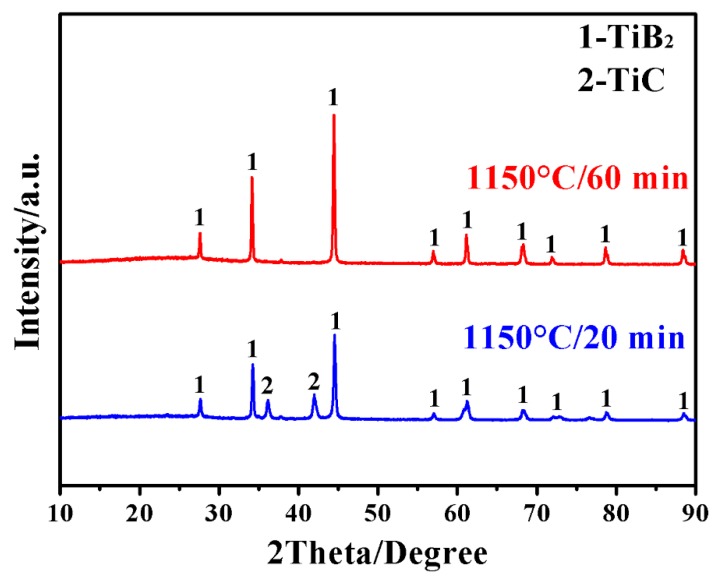
Phase formation in samples prepared by MSM-BCTR at 1150 °C for respectively 20 and 60 min.

**Figure 2 materials-12-02555-f002:**
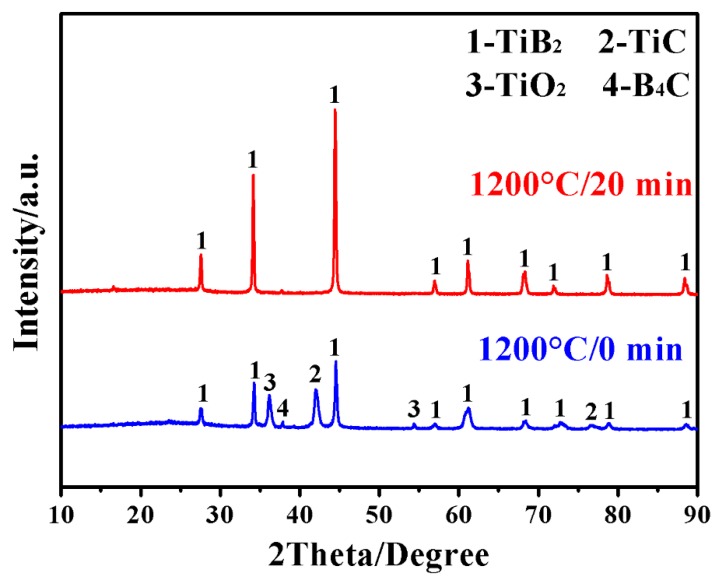
Phase formation in samples after MSM-BCTR treatments at 1200 °C for 0 and 20 min, respectively.

**Figure 3 materials-12-02555-f003:**
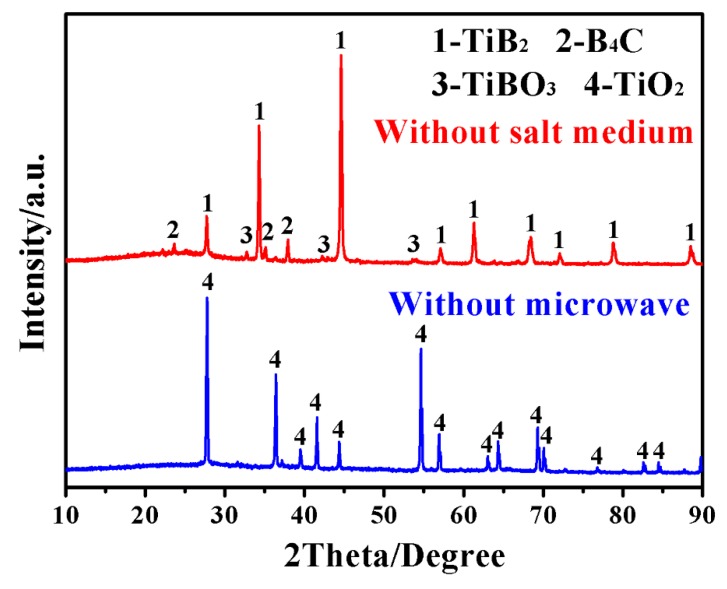
Phase formation in MSM-BCTR samples resultant from 20 min firing at 1200 °C, using solely microwave heating or molten-salt medium.

**Figure 4 materials-12-02555-f004:**
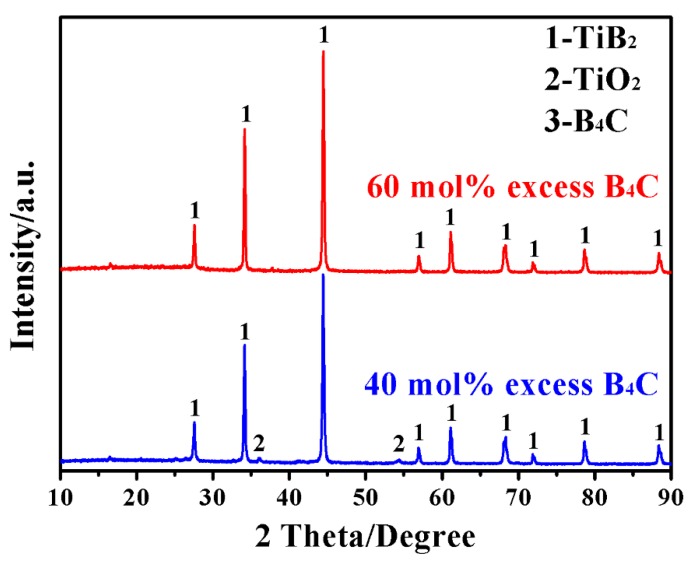
XRD patterns of samples containing initially 40 and 60 mol% excessive B_4_C, after MSM-BCTR treatments at 1200 °C for 20 min.

**Figure 5 materials-12-02555-f005:**
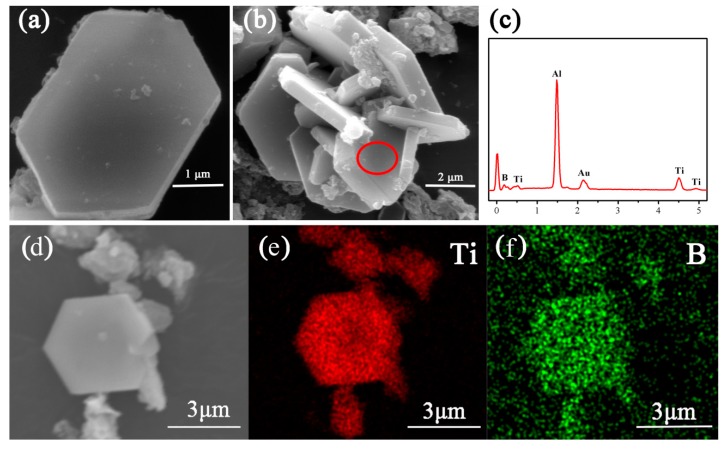
SEM images of (**a**) monodispersed TiB_2_ microplatelet and (**b**) interlaced TiB_2_ microplatelets, (**c**) corresponding EDS results of TiB_2_ microplatelets, (**d**) a lower magnification SEM image with EDS mappings of (**e**) Ti element and (**f**) B element prepared *via* MSM-BCTR at 1200 °C for 20 min. (Au and Al peaks in Figure 5b were from the gold coating and aluminum foil used for enhancing electrical conductivity of the powder sample.)

**Figure 6 materials-12-02555-f006:**
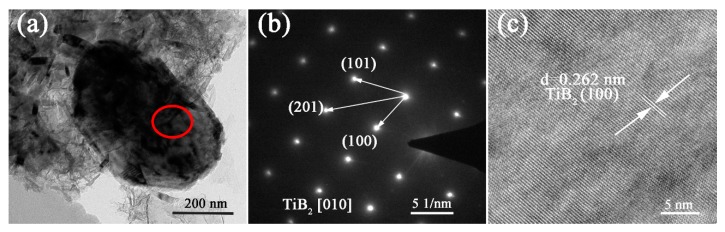
(**a**) Low-magnification TEM image and (**b**) SAED pattern of a representative TiB_2_ microplatelet in the sample prepared *via* MSM-BCTR at 1200 °C for 20 min, and (**c**) high-resolution TEM image taken in the area defined by the red circle in (**a**).

**Table 1 materials-12-02555-t001:** Batch Compositions and Processing Parameters for Manufacturing TiB_2_ Powders by the MSM-BCTR method.

Sample No.	Molar Ratio			Heating Mode	Temperature (°C)	Dwelling Time (min)	Salt Medium
	TiO_2_	B_4_C	C				
MSMBC-1	1.0	0.8	1.5	MWH †	1150	20	NaCl/KCl
MSMBC-2	1.0	0.8	1.5	MWH	1150	60	NaCl/KCl
MSMBC-3	1.0	0.8	1.5	MWH	1200	0	NaCl/KCl
MSMBC-4	1.0	0.8	1.5	MWH	1200	20	NaCl/KCl
MSMBC-5	1.0	0.8	1.5	CH†	1200	20	NaCl/KCl
MSMBC-6	1.0	0.8	1.5	MWH	1200	20	–
MSMBC-7	1.0	0.7	1.5	MWH	1200	20	NaCl/KCl

† MWH and CH denote microwave heating and conventional heating, respectively.

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
