# Peer review of "Low-Temperature High-Efficiency Preparation of TiB2 Micro-Platelets via Boro/Carbothermal Reduction in Microwave Heated Molten Salt"

_materials, 2019, doi:10.3390/ma12162555_

Round 1

Reviewer 1 Report

J. Liu et al. present a manuscript that studies the preparation of TiB2 micro-platelets via boro/carbothermal reduction in microwave heated molten salt. Experimentally the authors show that the using of the developed procedure resulted in the formation of hexagonal TiB2 microplatelets the same while using conventional reduction routes. The authors specify the effectiveness of microwave heating and molten-salt medium on the synthesis of TiB2 whilst significantly reducing the synthesis temperature towards 1150 – 1200 oC. 

The manuscript could be interesting to the readership of Materials once the following points have been addressed:

1.       What do the authors understand under high-efficiency preparation? If the high efficiency is reached due to a low temperature, you can skip high-efficiency in the title of the paper and in the abstract.

2.       Some references are missed in the text and should be added.

a.       Introduction, the line 56-58….” Apart from reducing the synthesis temperature and accelerating the reaction process, it facilitates diffusive mass transfer and self-assembly processes and thus the formation of the highly-crystalline product with intrinsically-anisotropic morphology “. Please put the relevant reference.

b.       Lines 215…217 “The high-resolution TEM image is shown in Figure 5(c) further reveals that the hexagonal micro-platelet had well-aligned lattice fringes with a constant interplanar spacing of 0.262 nm, which was in good agreement with the (100) interplanar distance of TiB2 crystal”. Please put the relevant reference.

c.       Lines 145…148 “(1150°C/60 min and 1200°C/20 min) in the present work were considerably milder than in the cases of conventional BCTR (up to 1600°C for several hours) using the identical type of reducing agent”.. Put a right reference… “and the molten-salt-solely-assisted BCTR (dwelling time of 4 h) using even much more expensive magnesium (Mg) as a reducing agent [16]” Put the synthesis temperature in the bracket.

3.       The authors said in the Method section that ….”Morphologies and sizes of TiB2 product powders were examined using a field emission microscope”. But in the text authors didn’t discuss the size of the powder and didn’t provide readers with the values of produced powders size.  This should be fixed!

4.       The TEM samples preparation procedure should be described in the Methods.

5.       Please, increase the resolution of the XRD patterns. It is difficult to distinguish small peaks.

6.       The main comments are related to the synthesis reactions and mechanism. The authors should describe in details the reaction (4) and (5). For readers would be difficult to understand the origin of B2O3, because according to methods section the synthesis is carried out under Ar atmosphere.

7.       The authors prove that in the case of using 40mol.% of excess B4C, despite the presence of the main phase of TiB2 and the absence of reducing agent, minor TiO2 was still detectable. This statement was based on the XRD pattern (Fig. 4). But, there is one low-intensive peak of TiO2 on the XRD pattern. It’s not enough to conclude that TiO2 is in the phase composition.

8.       The authors say…”Based on the comparison, it could be concluded that, under the present MSM-BCTR conditions, 60  mol% excessive B4C should be used for synthesizing phase pure TiB2 powder with the least impurity”. It can either be phase pure or contains some impurities, but not both. At all, what impurities do the authors mean?

9.       The authors should discuss the size and size distribution of the produced powder.

10.   Some typos are in the text….

Line 87… “ the amount of B4C used was…”

Line 110…” No.03-035-0798 and No.03-065-1118 are used”. Authors used the past tense in the manuscript. Correct the sentence according to past tense.

Author Response

Response to Reviewer’s Comments

Dear Editor of “Materials”:

Thank you very much for your consideration of our as-submitted manuscript entitled “Low-Temperature High-Efficiency Preparation of TiB2 Micro-Platelets via Boro/Carbothermal Reduction in Microwave Heated Molten Salt” for publication. We would also like to express our sincere gratitudes to you and the reviewer for your valuable comments. According to the comments and suggestions, we carefully revised our manuscript, and detailedly responded to each comment in this letter. We hope that our responses are satisfactory and the revised manuscript can be accepted for publication. If you have any further question, please do not hesitate to contact us.

It should be noted that, the modified words and sentences in the revised manuscript were highlighted with red color, and all the comments were sequentially responded as follows.

To Reviewer 1:

Comment (1):

What do the authors understand under high-efficiency preparation? If the high efficiency is reached due to a low temperature, you can skip high-efficiency in the title of the paper and in the abstract.

Answer to comment (1):

Thanks for your reasonable comment!

To the best of our knowledge, rising temperature has a common positive effect on enhancing the efficiencies of all the known chemical reactions. Therefore, it is a simple but effective way of accelerating chemical reactions by elevating processing temperature, although the energy consumption would increase accordingly. However, for the purpose of developing an advanced method for the industrial preparation of TiB2 powders, not only high efficiency but also low temperature were desirable for reducing the energy and time consumptions as well as the resulting preparation cost. But unfortunately, the methods reported by literatures for synthesizing TiB2 had to improve their efficiencies at the expense of increasing processing temperature. As a result, it is favorable for our present method to acquire high efficiency for synthesizing TiB2 even at reduced temperatures, and both of high-efficiency and low temperature were the key innovations of this paper.

Consequently, we believe that it is more appropriate to retain the phrase of “high-efficiency” in the sections of title and abstract.

Comment (2):

Some references are missed in the text and should be added.

Introduction, the line 56-58….” Apart from reducing the synthesis temperature and accelerating the reaction process, it facilitates diffusive mass transfer and self-assembly processes and thus the formation of the highly-crystalline product with intrinsically-anisotropic morphology “. Please put the relevant reference. Lines 215…217 “The high-resolution TEM image is shown in Figure 5(c) further reveals that the hexagonal micro-platelet had well-aligned lattice fringes with a constant interplanar spacing of 0.262 nm, which was in good agreement with the (100) interplanar distance of TiB2 crystal”. Please put the relevant reference. Lines 145…148 “(1150°C/60 min and 1200°C/20 min) in the present work were considerably milder than in the cases of conventional BCTR (up to 1600°C for several hours) using the identical type of reducing agent”.. Put a right reference… “and the molten-salt-solely-assisted BCTR (dwelling time of 4 h) using even much more expensive magnesium (Mg) as a reducing agent [16]” Put the synthesis temperature in the bracket.

Answer to comment (2):

Thanks very much for the comment!

According to the comment, we added some relevant references into the designated sections of our revised manuscript. (Line 21-25 Page 2, Line 4 Page 12, Line 9-10 Page7, respectively.)

Comment (3):

The authors said in the Method section that ….”Morphologies and sizes of TiB2 product powders were examined using a field emission microscope”. But in the text authors didn’t discuss the size of the powder and didn’t provide readers with the values of produced powders size. This should be fixed!

Answer to comment (3):

Thanks for the comments!

According to your comment, we discussed the accurate size of produced powders in the revised manuscript as follows, “And the highly-crystallized hexagonal TiB2 micro-platelets generally had micron-scale lengths (average length of 4.2 µm) and submicron-scale thicknesses (average thickness of 0.6 µm) with narrow size distributions.” (Line 8-11, Page 10 )

Comment (4):

The TEM samples preparation procedure should be described in the Methods.

Answer to comment (4):

Thanks for your suggestion!

According to your comment, we added the description of the preparation procedure of TEM samples into the Methods section of the resubmitted manuscript as follows, “For TEM characterization, the as-obtained powders were firstly ultrasonically dispersed in ethanol for 30 min, followed by natural seasoning in the air.” (Line 9-11, Page 5)

Comment (5):

Please, increase the resolution of the XRD patterns. It is difficult to distinguish small peaks.

Answer to comment (5):

Thanks for your comment!

We increased the resolution and quality of all the XRD patterns (Figure 1, Figure 2, Figure 3 and Figure 4) to distinguish small peaks and embedded them into the revised manuscript.

Comment (6):

The main comments are related to the synthesis reactions and mechanism. The authors should describe in details the reaction (4) and (5). For readers would be difficult to understand the origin of B2O3, because according to methods section the synthesis is carried out under Ar atmosphere.

Answer to comment (6):

Thanks very much for your comment!

According to your comment, we detailedly described the Reaction (4-5) as follows:

“due to microwave-assisted BCTR under identical processing conditions, the XRD pattern of sample (sample MSMBC-6) showed the peaks of TiB2, B4C and TiBO3, in which the intensities of TiB2 peaks were obviously higher than that of others, indicating the higher crystallization degree of TiB2 phase. Apart from TiB2, un-reacted B4C were still present along with the intermediate phase of TiBO3 which was formed at the temperature of 600°C via Reaction (4) but tended to be further converted into TiB2 under the condition of excess B4C (via Reaction (5)) [13]. It should be noted that, in this process, the requirement of excess boron source was mainly attributable to the evaporation loss of the intermediate product of B2O3. Specifically, owing to its low melting point and high volatility, B2O3, as the product of borothermal reduction reaction (Reaction (6)) would suffer from significant evaporation loss even at a low temperature range of 900-1100°C, thus resulting in incompletion of the subsequent carbothermal reduction reaction (Reaction (7)) for synthesizing TiB2 [21].”

TiO2(s) +B2O3(l,g) +C(s)=2TiBO3(s) +CO (g)                                   (4)

2TiBO3(s) +B2O3(l,g) +9C(s)=2TiB2(s) +9CO (g)                               (5)

TiO2(s)+B4C(s)=TiB2(s)+B2O3(l,g)+CO(g)                             (6)

TiO2(s)+B2O3(l,g)+C(s)=TiB2(s)+CO(g)                                 (7)

Comment (7):

The authors prove that in the case of using 40mol.% of excess B4C, despite the presence of the main phase of TiB2 and the absence of reducing agent, minor TiO2 was still detectable. This statement was based on the XRD pattern (Fig. 4). But, there is one low-intensive peak of TiO2 on the XRD pattern. It’s not enough to conclude that TiO2 is in the phase composition.

Answer to comment (7):

Thanks very much for your comment!

After we increasing the resolution and quality of Figure 4, one more diffraction peak with a low intensity was also observed. In addition to TiB2, the only two detectable diffraction peaks with the 2Theta locations of 36.053° and 54.362° were respectively indexed to the (101) and (211) crystallographic planes of TiO2, which have the highest diffraction intensities in the standard X-ray diffraction pattern (ICDD cards No.03-065-1118). Therefore, we inferred that the impurity of the powder product was mainly composed of TiO2, and the ultra-low intensities of TiO2 phase in corresponding XRD pattern was attributable to its low weight content.

Comment (8):

The authors say…”Based on the comparison, it could be concluded that, under the present MSM-BCTR conditions, 60 mol% excessive B4C should be used for synthesizing phase pure TiB2 powder with the least impurity”. It can either be phase pure or contains some impurities, but not both. At all, what impurities do the authors mean?

Answer to comment (8):

Thanks very much for your comment!

We are very sorry that we aroused misunderstanding of readers. Actually, there was no detectable impurity phase in the XRD pattern of as-obtained TiB2 powders. Therefore, this sentence was rewritten as follows, “Based on the comparison, it could be concluded that, under the present MSM-BCTR conditions, 60 mol% excessive B4C should be used for synthesizing phase pure TiB2 powder ”. (Line 20-22, Page 9)

Comment (9):

The authors should discuss the size and size distribution of the produced powder.

Answer to comment (9):

Thanks very much for your comment!

We discussed the size and size distribution of the produced powder in the revised manuscript as follows, “And the highly-crystallized hexagonal TiB2 micro-platelets generally had micron-sized lengths (average length of 4.2 µm) and submicron-scale thicknesses (average thickness of 0.6 µm) with narrow size distributions.” (Line 8-11, Page 10 )

Comment (10):

Some typos are in the text…. Line 87… “ the amount of B4C used was…” Line 110…” No.03-035-0798 and No.03-065-1118 are used”. Authors used the past tense in the manuscript. Correct the sentence according to past tense.

Answer to comment (10):

Thanks very much for your comment!

According to your comment, we corrected the sentence as “To compensate for the volatilization loss of boron species at high temperature [30], the amount of B4C was slightly higher than the stoichiometric amount indicated by Reaction (1).” (Line 2, Page 4)

In the text, the tense of sentence was corrected to past tense. The revised sentence was as follows, “ICDD cards of No.00-035-0741, No.03-065-8805, No.03-035-0798 and No.03-065-1118 were used to identify TiB2, TiC, B4C and TiO2, respectively.” (Line 4, Page 5)

Thanks again for your valuable suggestions and commnets! We sincerely wished the revised manuscript could meet the standard for publication!

Best Regards!

Yours Sincerely,

Dr. Jianghao Liu, on behalf of all co-authors

Aug. 1st, 2019

Address: The State Key Laboratory of Refractories and Metallurgy, Wuhan University of Science and Technology, Wuhan 430081, China

*Corresponding author: Dr. Jianghao Liu, Email: [email protected]

Prof. Haijun Zhang, Email: [email protected]

Reviewer 2 Report

I suggest to accept the paper in the present form

Author Response

Response to Reviewer’s Comments

Dear Editor of “Materials”:

Thank you very much for your consideration of our as-submitted manuscript entitled “Low-Temperature High-Efficiency Preparation of TiB2 Micro-Platelets via Boro/Carbothermal Reduction in Microwave Heated Molten Salt” for publication. We would also like to express our sincere gratitudes to you and the reviewer for your valuable comments. According to the comments and suggestions, we carefully revised our manuscript, and detailedly responded to each comment in this letter. We hope that our responses are satisfactory and the revised manuscript can be accepted for publication. If you have any further question, please do not hesitate to contact us.

It should be noted that, the modified words and sentences in the revised manuscript were highlighted with red color, and all the comments were sequentially responded as follows.

To Reviewer 2:

Comment (1):

I suggest to accept the paper in the present form

Answer to comment (1):

Thanks very much for the comment of our paper for publication!

Best Regards!

Yours Sincerely,

Dr. Jianghao Liu, on behalf of all co-authors

Aug. 1st, 2019

Address: The State Key Laboratory of Refractories and Metallurgy, Wuhan University of Science and Technology, Wuhan 430081, China

*Corresponding author: Dr. Jianghao Liu, Email: [email protected]

Prof. Haijun Zhang, Email: [email protected]

Reviewer 3 Report

The figures appear to be out of focus. Please improve their size and quality before their publication.

Author Response

Response to Reviewer’s Comments

Dear Editor of “Materials”:

Thank you very much for your consideration of our as-submitted manuscript entitled “Low-Temperature High-Efficiency Preparation of TiB2 Micro-Platelets via Boro/Carbothermal Reduction in Microwave Heated Molten Salt” for publication. We would also like to express our sincere gratitudes to you and the reviewer for your valuable comments. According to the comments and suggestions, we carefully revised our manuscript, and detailedly responded to each comment in this letter. We hope that our responses are satisfactory and the revised manuscript can be accepted for publication. If you have any further question, please do not hesitate to contact us.

It should be noted that, the modified words and sentences in the revised manuscript were highlighted with red color, and all the comments were sequentially responded as follows.

To Reviewer 3:

Comment (1):

The figures appear to be out of focus. Please improve their size and quality before their publication.

Answer to comment (1):

Thanks very much for your comment!

We improved the resolution and quality as well as size of all figures in the resubmitted manuscript.

Thanks very much for the valuable suggestions and consideration of our paper for publication!

We have carefully revised the paper according to the comments of reviewer, and we hoped it could meet the standard of “Materials”.

Best Regards!

Yours Sincerely,

Dr. Jianghao Liu, on behalf of all co-authors

Aug. 1st, 2019

Address: The State Key Laboratory of Refractories and Metallurgy, Wuhan University of Science and Technology, Wuhan 430081, China

*Corresponding author: Dr. Jianghao Liu, Email: [email protected]

Prof. Haijun Zhang, Email: [email protected]

Round 2

Reviewer 1 Report

ACCEPT FOR PUBLICATION

Reviewer 3 Report

N/A

This manuscript is a resubmission of an earlier submission. The following is a list of the peer review reports and author responses from that submission.